# FREQUENT SUBGRAPH-BASED PERSISTENT HOMOLOGY FOR GRAPH CLASSIFICATION

## ABSTRACT

Persistent homology (PH) has recently emerged as a powerful tool for extracting topological features. Integrating PH into both machine learning and deep learning models enhances their topology-awareness and interpretability. However, most PH methods on graphs rely on a limited set of filtrations (e.g., degree- or weight-based), which overlook richer features such as recurring information across the dataset, thereby restricting their expressive power. In this work, we propose a novel filtration on graphs, called *Frequent Subgraph Filtration* (FSF), which is derived from frequent subgraphs and produces stable and information-rich *Frequency-based Persistent Homology* (FPH) features. We explore the theoretical properties of FSF and provide proofs and experimental validation of them. Beyond persistent homology itself, we further introduce two approaches for graph classification: (i) an FPH-based machine learning model (FPH-ML), and (ii) a hybrid framework integrating FPH with graph neural networks (FPH-GNNs) to enhance topology-aware graph representation learning. Our proposed frameworks show the potential of bridging frequent subgraph mining and topological data analysis, offering a new perspective on topology-aware feature extraction and graph representation learning. Experimental results show that FPH-ML achieves competitive or superior accuracy compared to kernel-based and degree-based filtration methods. When injected into GNNs, FPH delivers relative gains of 0.4–21% (up to +8.2 pts) over their GCN/GIN backbones across benchmarks.

## 1 INTRODUCTION

Graph classification is fundamental to graph learning, with applications in social networks, cheminformatics, and bioinformatics (Liu et al., 2023). Existing approaches fall into two main categories: kernel-based and graph neural network (GNN)-based methods (Aktas et al., 2019). Kernel methods, including random walk (Kashima et al., 2004), graphlet (Shervashidze et al., 2009), and Weisfeiler–Lehman (WL) kernels (Shervashidze et al., 2011), offer strong theoretical foundations but scale poorly to large datasets. In contrast, GNNs have emerged as the dominant paradigm, learning graph representations end-to-end and achieving state-of-the-art performance on diverse benchmarks (Ju et al., 2025; Shi et al., 2025). These models primarily focus on local neighborhood aggregation, effectively capturing local structural information. However, despite these advances, GNNs often struggle to capture global, high-order topological features, which are crucial for distinguishing graphs, especially in domains where topological structure plays a central role.

To overcome these limitations, there has been growing interest in integrating topological data analysis (TDA) with machine learning and GNNs, as graphs are inherently topological structures (Immonen et al., 2023; Pun et al., 2018). Among TDA techniques, Persistent Homology (PH) is a powerful method for extracting multi-dimensional topological features, such as connected components, cycles, and voids. PH tracks the birth and death of these features across a filtration sequence, resulting in persistence diagrams that provide a compact and informative summary of the graph's topological structure. Unlike GNNs, which primarily focus on local neighbourhood information, PH is capable of capturing global, high-dimensional topological information, offering complementary insights for graph representation and enhancing the interpretability of models.

A key component of PH is the filtration function, which defines the order in which graph elements (nodes or edges) are added to construct a sequence of nested simplicial complexes. Most exist-

ing graph filtrations are primarily based on edge weights (e.g., Vietoris-Rips (Petri et al., 2013), Dowker sink and source filtrations (Chowdhury & Mémoli, 2018)), which limits their applicability to weighted graphs. Other approaches, such as degree-based filtrations, rely on simple local properties of vertices and often fail to capture meaningful global topological structures (Aktas et al., 2019). Therefore, designing more expressive and diverse filtration strategies is essential for enhancing the representational capacity of PH in graph learning tasks.

To address these limitations, we propose a novel graph filtration based on frequent subgraph patterns. Our method mines limited-size frequent patterns across the dataset and constructs a filtration for each graph via isomorphic mapping. This design incorporates recurring structural information and captures global dataset-level topology. We integrate the filtration into both traditional machine learning and GNN pipelines for graph classification. Our main contributions are:

(i) We propose *Frequent Subgraph Filtration* (FSF), the first frequency-driven filtration for graphs, which integrates frequent subgraph patterns information into persistent homology (PH) to capture recurring, stable, and dataset-level topological information.

(ii) We explore and prove the theoretical properties of FSF, including the PH dimension (bounded by pattern size), monotonicity and isomorphism invariance.

(iii) We introduce two graph classification approaches: (a) a *Frequency-based Persistent Homology* (FPH) machine learning model (FPH-ML), and (b) a hybrid framework integrating FPH with graph neural networks (FPH-GNN) to enhance topology-aware graph representation learning.

(iv) Extensive experiments on graph classification benchmarks demonstrate that our methods consistently outperform kernel-based baselines, existing PH-based models, and GNNs on most datasets.

## 2 RELATED WORK

**Frequent subgraph mining.** Frequent subgraph mining (FSM) is a fundamental graph mining task that discovers recurring substructures in datasets (Rehman et al., 2024). Early methods, such as AGM (Inokuchi et al., 2000), follow an Apriori-style candidate generation, but suffer from high costs of repeated subgraph isomorphism tests. To overcome this, gSpan (Yan & Han, 2002) adopts DFS codes with a depth-first extension strategy, while Grami (Elseidy et al., 2014) encodes subgraph isomorphism as a constraint satisfaction problem for efficient growth and pruning.

Since not all frequent subgraphs are meaningful, methods like cgSpan (Shaul & Naaz, 2021) and FCSG (Chen et al., 2024) focus on concise representations. For dynamic graphs, DyFSM (Chen et al., 2023) uses a fringe set to incrementally update frequent subgraphs as databases evolve, while He et al. (He et al., 2025; 2024) propose frameworks for mining sequences in dynamic attributed graphs and extracting credible attribute rules. FSM has also been extended beyond simple graphs. for example, Aslay et al. (Aslay et al., 2018) study hypergraphs, and FreSCo (Preti et al., 2022) generalizes FSM to simplicial complexes. More recently, learning-based FSM has attracted increasing attention. SPMiner (Ying et al., 2024) is the first approach to represent subgraphs in a learned embedding space, leveraging order embeddings to preserve the hierarchical relation between subgraphs and their supergraphs. Multi-SPMiner (Zeghina et al., 2023) extends this framework to multi-graph settings.

**Persistent homology and its application.** Topological data analysis (TDA) has gained increasing attention, with persistent homology (PH) as its most widely used tool for tracking topological features across scales (Fugacci et al., 2016). PH quantifies structures such as connected components, loops, and voids (Edelsbrunner et al., 2008), typically summarized in persistence diagrams that record feature birth and death times. To facilitate interpretation, other representations have been developed, including barcodes (Wadhwa et al., 2018), persistence landscapes (Flammer & Hüper, 2024), and persistence images (Adams et al., 2017).

PH has been applied in diverse domains, including neuroscience (Liang et al., 2021), GIScience (Corcoran & Jones, 2023b), and time-series analysis (Ravishanker & Chen, 2021). For example, Corcoran and Jones (Corcoran & Jones, 2023a) apply PH to GIScience, demonstrating robustness, void detection, and downstream analysis through PH-based signatures. Beyond direct applications, PH has been integrated into machine learning (Pun et al., 2018) and increasingly combined with deep learning (Pham et al., 2025). Persistent homology can enhance neural networks by providing

higher-order structural information. For example, Zhao et al. (Zhao et al., 2020) proposed a novel network architecture where PH guides graph neural networks (GNNs) by reweighting message passing between graph nodes. Similarly, Rephine (Immonen et al., 2023) is a framework that extends PH by designing a new refined filtration, enabling richer topological features to be incorporated into graph representation.

**Graph classification.** Graph classification aims to assign labels to entire graphs. Research has evolved from kernel-based similarity measures to embedding methods and, more recently, expressive and scalable GNNs. Early work focused on graph kernels combined with classifiers such as SVMs (Kriege et al., 2020). Representative examples include Random Walk Kernels (Kashima et al., 2004), which count matching random walks; Graphlet Kernels (Shervashidze et al., 2009), which compare distributions of small induced subgraphs; and Weisfeiler–Lehman (WL) Kernels (Shervashidze et al., 2011), which iteratively relabel neighborhoods to capture richer structures. These methods perform well on small and medium datasets but suffer from high computational cost and poor scalability. To overcome these issues, graph embedding methods learn low-dimensional vector representations without explicit pairwise kernel computation, e.g., Graph2Vec (Narayanan et al., 2017) provides fixed-size embeddings for entire graphs.

Graph Neural Networks (GNNs) have recently become the dominant approach for graph classification. They follow the message-passing paradigm, where nodes aggregate information from neighbors and a readout operation produces graph-level embeddings (Gilmer et al., 2017). Representative models include GCNs (Kipf & Welling, 2016), which apply spectral filtering; GraphSAGE (Hamilton et al., 2017), which uses neighborhood sampling for inductive learning; GATs (Veličković et al., 2017), which incorporate self-attention; and GINs (Xu et al., 2019), which achieve Weisfeiler–Lehman-level discriminative power through expressive aggregation.

## 3 PRELIMINARIES

**Frequent Subgraph mining.** A vertex-labeled undirected graph is a tuple $G = (V, E, \ell)$, where $V$ is a finite set of vertices, $E \subseteq \{\{u, v\} \mid u, v \in V, u \neq v\}$ is a set of undirected edges, and $\ell : V \to L$ is a label function mapping each vertex to a label from a finite label set $L$. Given two vertex-labeled undirected graphs $G = (V, E, \ell)$ and $G' = (V', E', \ell')$. If there exists a bijection $\phi : V \to V'$ such that: $(u, v) \in E \Leftrightarrow (\phi(u), \phi(v)) \in E'$, $\ell(u) = \ell'(\phi(u))$ for all $u \in V$, then $G$ and $G'$ are isomorphic, denoted as $G \cong G'$. Then, we give the definition of MNI-frequent subgraph.

**Definition 1 (MNI-Frequent Subgraph)** *Let $\mathcal{D} = (V, E, \ell)$ be a single vertex-labeled undirected graph, and let $g = (V_g, E_g, \ell_g)$ be a vertex-labeled subgraph. An embedding of $g$ in $\mathcal{D}$ is an injective mapping $\phi : V_g \to V$ that preserves adjacency and vertex labels. We denote by $\mathrm{Emb}(g, \mathcal{D})$ the set of all such embeddings. For each vertex $u \in V_g$, define the distinct node-image set $\mathrm{Img}(u) := \{ \phi(u) \mid \phi \in \mathrm{Emb}(g, \mathcal{D}) \}$. The Minimum Node Image (MNI) support of $g$ in $\mathcal{D}$ is*

$$\mathrm{MNI}(g, \mathcal{D}) = \min_{u \in V_g} \big|\mathrm{Img}(u)\big|.$$

*Then $g$ is called* MNI-frequent *if $MNI(g, \mathcal{D}) \geq \sigma$, where $\sigma \in \mathbb{N}$ is a user-defined minimum support threshold.*

**Simplicial complex.** Let $V = \{v_0, v_1, \ldots, v_k\}$ be a set of $k + 1$ affinely independent points. Then, their convex hull is defined as a $k$-simplex, where $k$ is the dimension of the simplex: $\sigma = \{x \in \mathbb{R}^n \mid x = \sum_{i=0}^{k} \lambda_i v_i, \lambda_i \geq 0, \quad \sum_{i=0}^{k} \lambda_i = 1\}$. Let $\mathcal{K}$ be the collection of simplices. If $\mathcal{K}$ satisfies the following two conditions, it is called a simplicial complex: i) Closure Condition: If $\sigma \in \mathcal{K}$ and $\tau$ is any face of $\sigma$, then $\forall \sigma \in \mathcal{K}, \forall \tau \subseteq \sigma, \tau \in \mathcal{K}$; and ii) Intersection Condition: If $\sigma, \sigma' \in \mathcal{K}$, then $\forall \sigma, \sigma' \in \mathcal{K}, \sigma \cap \sigma' \in \mathcal{K}$ or $\sigma \cap \sigma' = \emptyset$.

**Homology.** Given a simplicial complex $\mathcal{K}$, the $k$-boundary group is $B_k(\mathcal{K}) = \mathrm{Im}(\partial_{k+1}) = \{\partial_{k+1}(\sigma) \mid \sigma \in C_{k+1}\}$, and the $k$-cycle group is $Z_k(\mathcal{K}) = \ker(\partial_k) = \{c \in C_k \mid \partial_k(c) = 0\}$. Since $B_k \subseteq Z_k \subseteq C_k$, cycles are $k$-chains with zero boundary, while boundaries are cycles arising from higher-dimensional chains. Homology identifies nontrivial cycles (not boundaries), yielding the $k$-th homology group

$$H_k(\mathcal{K}) = Z_k(\mathcal{K})/B_k(\mathcal{K}),$$

whose rank, the Betti number $\beta_k = \dim H_k(\mathcal{K})$, counts the independent $k$-dimensional topological features.

**Definition 2 (Filtration Sequence)** *Given a simplicial complex $\mathcal{K}$, a filtration sequence is a nested sequence of sub-simplicial complexes: $\emptyset = \mathcal{K}_{\epsilon_0} \subseteq \mathcal{K}_{\epsilon_1} \subseteq \mathcal{K}_{\epsilon_2} \subseteq \cdots \subseteq \mathcal{K}_{\epsilon_n} = \mathcal{K}$. Each $\mathcal{K}_{\epsilon_i}$ is a subcomplex of $\mathcal{K}$, and as the scale parameter $\epsilon$ increases, the simplicial complex grows. That is,*

$$\epsilon_0 \leq \epsilon_1 \leq \cdots \leq \epsilon_{n-1} \leq \epsilon_n.$$

Thus, we can introduce the definition of persistent homology:

**Definition 3 (Persistent Homology)** *Let $\mathcal{K}$ be a simplicial complex with a filtration sequence $[\mathcal{K}_{\epsilon_0} \subseteq \mathcal{K}_{\epsilon_1} \subseteq \mathcal{K}_{\epsilon_2} \subseteq \cdots \subseteq \mathcal{K}_{\epsilon_n}]$. For each $\mathcal{K}_{\epsilon_i} \in \mathcal{K}$, the corresponding $k$-th homology group is $H_k(\mathcal{K}_{\epsilon_i})$. The birth of a topological feature $\alpha$ is defined as its first appearance in $\mathcal{K}_{\epsilon_b}$, and its death is when it disappears in $\mathcal{K}_{\epsilon_d}$. The persistent homology group is then defined as:*

$$H_k^{b,d} = \text{Im} \left( H_k(\mathcal{K}_{\epsilon_b}) \to H_k(\mathcal{K}_{\epsilon_d}) \right).$$

## 4    FREQUENT SUBGRAPH-BASED FILTRATION

In this paper, we propose a novel filtration method constructed from frequent subgraph patterns. The framework for filtration construction is shown in Figure 7 in Appendix A.3. Given a vertex-labeled graph transaction dataset $\mathcal{D} = \{G_0, G_1, \ldots, G_n\}$, we consider the union of all graphs as a large graph $\mathcal{G}$, which serves as the input for FSM. We now formalize the $k$-size Frequent Subgraph Mining (k-FSM) problem.

**Problem (k-size Frequent Subgraph Mining (k-FSM))**
Given a user-defined minimum support threshold $\sigma \in \mathbb{N}$ and a maximum subgraph size $k$ (where size refers to the number of vertices in the subgraph), the goal of k-FSM is to discover all subgraphs of size at most $k$ that are MNI frequent subgraph (Definition 1) in $\mathcal{G}$. Formally, a subgraph $g$ is considered k-size frequent if:

$$\mathcal{F}_k(\mathcal{G}, \sigma) = \{g \in \mathcal{G}_k \mid \text{MNI}(g, \mathcal{G}) \geq \sigma, \; |V_g| \leq k\}.$$

We perform k-FSM on $\mathcal{G}$ to obtain a collection of frequent subgraph patterns, denoted as $P_{ij}$, where $i$ indexes the subgraph pattern and $j$ corresponds to the frequency threshold level. Subsequently, for each graph $G_i$, we extract the set of vertices that participate in the isomorphisms of frequent subgraph patterns. These vertex sets are then treated as simplices to construct a simplicial complex for $G_i$. This complex encodes high-order topological relationships among frequently co-occurring vertex groups, thus capturing richer structural information. Finally, we define a filtration by progressively adding simplices to the complex based on the decreasing order of their corresponding subgraph frequency. This process results in a k-FSF sequence for each graph $G_i$.

**Definition 4 (k-size Frequent Subgraph-Based Filtration (k-FSF))** *Let $\mathcal{D}$ be a graph transaction dataset, and let $G \in \mathcal{D}$ be a graph. Let $P_j = \{p_{ij}\}$ denote the set of frequent subgraph patterns at the $j$-th frequency threshold, where each $p_{ij}$ is a subgraph pattern identified via k-FSM.*

*For a filtration value $t \in \mathbb{R}_+$, we define the simplicial complex $\mathcal{K}_t$ on $G$ as:*

$$\mathcal{K}_t = \left\{ \triangle(T) \; \middle| \; \exists \, p_{ij} \subseteq G, \; MNI(p_{ij}, \mathcal{D}) \geq \frac{1}{t}, \; \text{and } T \subseteq I(V_{p_{ij}}) \right\}$$

*where $\frac{1}{t}$ corresponds to the frequency threshold for subgraph patterns. $\triangle(T)$ denotes a simplex formed by the vertex set $T$. $V_{p_{ij}}$ is the vertex set of the subgraph pattern $p_{ij}$, and $I(V_{p_{ij}}, \mathcal{G})$ is the set of vertex images of $V_{p_{ij}}$ under subgraph isomorphisms from $p_{ij}$ to $G$.*

*The filtration sequence is then defined as:*

$$\{\mathcal{K}_t\}_{t \in \mathbb{R}_+}.$$

*As $t$ increases (i.e., as $\frac{1}{t}$ decreases), more simplices are progressively added to the complex, thereby enlarging the topological structure.*

For example, figure 1 illustrates an example $k$-FSF $\{\mathcal{K}_t\}_{t_0 < t_1 < t_2}$. Subfigures 1a–c show the simplicial complexes at times $t_0$, $t_1$, and $t_2$, respectively. As $t$ increases, additional simplices are incorporated, enriching the topology. For example, the isolated component $[730, 731]$ at $t_0$ becomes connected to the main complex at $t_1$ through 1-simplices $[713, 730]$, $[714, 730]$, and $[713, 731]$, as well as a 2-simplex $[713, 714, 716]$. At $t_2$, further 2-simplices, such as $[713, 730, 716]$, are added, increasing topological complexity.

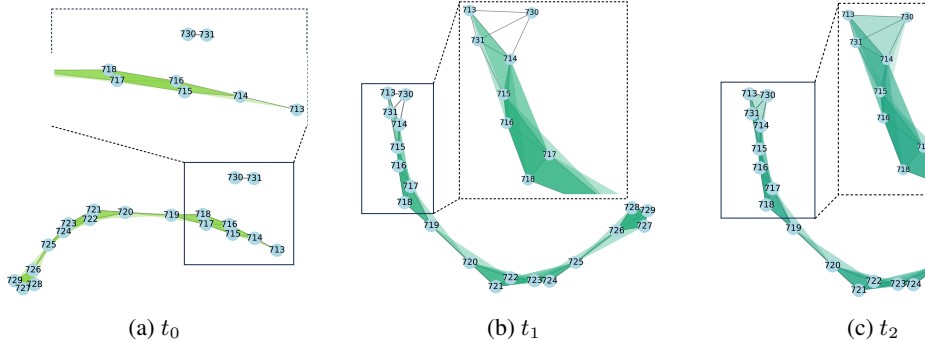

(a) $t_0$           (b) $t_1$           (c) $t_2$

Figure 1: Example of k-FSF.

Here, we give three propositions on FSF, and the formal proofs are provided in Appendix A.2.

**Proposition 1 (Homology Dimension of k-FSF)** *Let $\{\mathcal{K}_t\}_{t \in \mathbb{R}_+}$ be a k-FSF on a graph $G$. Then the maximum dimension $\mathrm{maxD}$ in which persistent homology can be nontrivial is bounded by:*

$$\mathrm{maxD} = \max\{p \in \mathbb{N} \mid H_p(\mathcal{K}_t) \neq 0\} \leq k - 1.$$

*Moreover, for $t_1 \leq t_2$, the persistent map $H_{k-1}(\mathcal{K}_{t_1}) \to H_{k-1}(\mathcal{K}_{t_2})$ is injective.*

Note that even for homology groups other than $H_{k-1}$, the groups $H_i(\mathcal{K}_t)$ need not vanish for $t \gg 1$. This contrasts with what happens in filtrations such as Vietoris-Rips (see A.2).

This shows that persistence obtained from FSF is not only a consequence of the frequency of patterns but also of the topology of the graph.

**Proposition 2 (Monotonicity of FSF)** *Let $\{\mathcal{K}_t\}_{t \in \mathbb{R}_+}$ be k-FSF constructed on a graph $G$. Then the filtration is monotonic with respect to $t$, i.e.,*

$$\mathcal{K}_{t_1} \subseteq \mathcal{K}_{t_2} \subseteq \cdots \subseteq \mathcal{K}_{t_n} = \mathcal{K} \quad \text{for } t_1 < t_2 < \cdots < t_n.$$

**Proposition 3 (FSF is Isomorphism Invariant)** *Let $G$ and $G'$ be two isomorphic vertex-labeled graphs, written $G \cong G'$. Then the FSFs constructed from $G$ and $G'$ are isomorphic; that is, for each filtration step $i$,*

$$\mathcal{K}_i \cong \mathcal{K}'_i,$$

*where $\mathcal{K}_i$ denotes the simplicial complex at filtration value $i$. Consequently, the persistence diagrams of $G$ and $G'$ are equal:*

$$D_G = D_{G'}.$$

## 5   FREQUENCY PERSISTENT HOMOLOGY FOR GRAPH CLASSIFICATION

In this section, we show how FPH features enhance graph classification through two approaches: (i) an FPH-based machine learning model, and (ii) integrating FPH with GNNs for enhancing topology-aware representation learning.

## 5.1 FPH-BASED MACHINE LEARNING.

Figure 2 illustrates the FPH-based machine learning (FPH-ML) pipeline. FPH features are extracted via the proposed $k$-FSF, vectorized into PH statistical features, and used as classifier inputs. Unlike graph kernels or GNNs, this approach relies solely on topological statistics from persistent homology, without kernels or message passing.

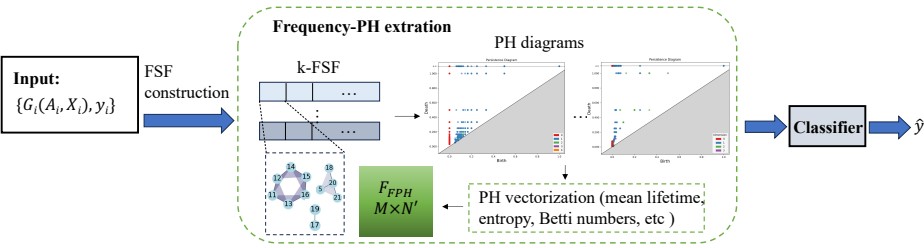

Figure 2: FPH-ML pipeline.

For each homology dimension $k \in \{0, \dots, d\}$ of graph $G_i$, let $\mathrm{PD}_i^k = \{(b_j^k, d_j^k)\}_{j=1}^{n_k}$ denote the persistence pairs, where infinite values are set to 1. The lifetime of each pair is $\ell_j^k = d_j^k - b_j^k$, and if $n_k = 0$ the features are zero-padded. From the lifetimes, we compute seven statistics: mean ($\mu$), maximum ($L_{\max}$), minimum ($L_{\min}$), median ($\tilde{L}$), standard deviation ($\sigma$), Betti number ($\beta$), and entropy ($E$). The total persistence across all finite pairs is $P_{\mathrm{tot},i} = \sum_{(b,d):\, d<\infty}(d-b)$. Concatenating features across all dimensions yields the PH vector

$$\mathbf{f}_i = \left[ \mu_i^0, L_{\max,i}^0, L_{\min,i}^0, \tilde{L}_i^0, \sigma_i^0, \beta_i^0, E_i^0, \dots, \mu_i^d, L_{\max,i}^d, L_{\min,i}^d, \tilde{L}_i^d, \sigma_i^d, \beta_i^d, E_i^d, P_{\mathrm{tot},i} \right].$$

We have $\mathbf{f}_i \in \mathbb{R}^{7(d+1)+1}$. Then, all the $f_i$ are stacked as the graph embedding with dimension $M \times N'$:

$$F_{\mathrm{FPH}} = [f_1; f_2; \dots; f_M] \in \mathbb{R}^{M \times N'}.$$

We then train a classifier $h : \mathbb{R}^{N'} \to \mathcal{Y}$ such that $\hat{y}_i = h(f_i)$.

## 5.2 COMBINING FPH WITH GNNS.

We develop FPH-GNN to enhance the expressiveness of GNN-based graph embeddings by integrating global, stable, and high-order topological information. Figure 3 illustrates the overall framework. Specifically, the FPH features are incorporated into the original graph input as a global PH token, guiding the GNN to be aware of global topological structure beyond local message passing. The input consists of a set of graphs $\{G_i(A_i, X_i, Y_i)\}_{i=1}^{M}$, where $A_i$ denotes the adjacency matrix, $X_i$ the node feature matrix, and $y_i$ the corresponding graph label. The key components of the framework are detailed below.

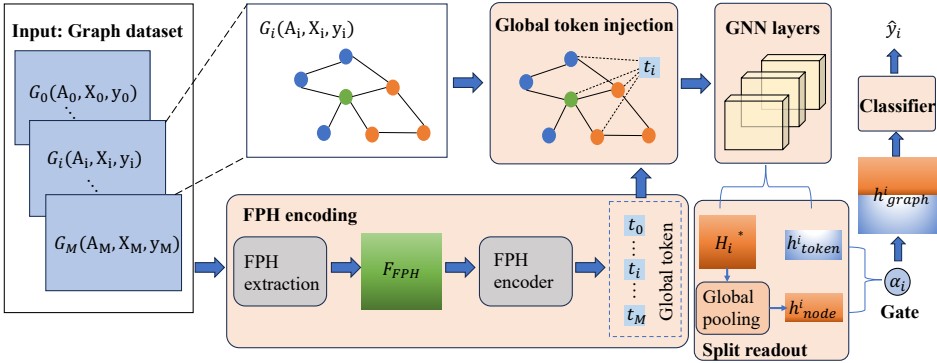

Figure 3: Overview of the proposed FPH-GNN framework.

**FPH encoding.** For each graph $G_i$, we precompute its topological feature $F_{\text{FPH}}^i \in \mathbb{R}^{d_{\text{ph}}}$. To align these topological features with the latent representation space of node embeddings, we employ a multi-layer perceptron (MLP) as an encoder. Then the FPH embedding is obtained:

$$e_{\text{fph}}^i = \text{MLP}\big(F_{\text{FPH}}^i\big) \in \mathbb{R}^H,$$

where $H$ denotes the hidden dimension.

**Global token injection.** The encoded FPH embedding $e_{\text{fph}}^i$ is injected into $G_i$ as a virtual global token node $t_i$ to serve as a global representative of topological knowledge. To maximize its influence, $t_i$ is connected to the top-$K$ highest-degree nodes in $G_i$, forming edges $(t_i, v)$ where $v \in \text{TopKDeg}(G_i)$. Consequently, $H_i^{(0)} \in \mathbb{R}^{n_i \times d}$ incorporates both node features and the injected FPH token, enabling subsequent GNN layers to jointly propagate global topological context alongside local structural information.

**GNN layers:** We use $L$ standard GNN layers to jointly propagate structural and FPH token information, with a residual connection from the initial embedding $H_i^{(0)}$ to the final representation $H_i^{(L)}$ to preserve topological signals (details in Appendix A.4).

**Split readout and gated fusion.** After $L$ GNN layers, we separately read out the representations of ordinary nodes and the FPH token. Since each graph has only one token node, its representation is $h_{\text{token}}^i = H_{i,t_i}^*$. For the ordinary nodes, we apply a permutation-invariant READOUT function (e.g., global sum pooling):

$$h_{\text{node}}^i = \text{READOUT}\big(\{H_{i,v}^* : v \in V_i\}\big).$$

To adaptively fuse structural and topological information, we compute a gate coefficient $\alpha_i$ via an MLP:

$$\alpha_i = \sigma\big(\text{MLP}_{\text{gate}}\big(F_{\text{FPH}}^i\big)\big), \quad h_{\text{graph}}^i = h_{\text{node}}^i + \alpha_i\, h_{\text{token}}^i.$$

The fused representation $h_{\text{graph}}^i$ is then fed to a classifier over $C$ classes, yielding $\hat{y}_i \in \mathbb{R}^C$.

Overall, FPH-GNN combines local structure captured by the GNN with global topological signals from FPH, enabling topology-aware graph representations.

## 6 EXPERIMENTS

We conduct a series of experiments to assess our proposed approach: (1) the robustness of FSF, (2) the performance of FPH-ML, which reveals the discriminative power of FPH features, and (3) the performance of the proposed FPH-GNN framework. Here, we set $k = 4$, which obtains the maximum PH dimension $H_3$ (Proposition 1). We consider $H_3$ (once born, never killed) is less informative. Thus, our FPH combines $H_0$, $H_1$, and $H_2$.

Table 1: Dataset statistics.

| Dataset | $|G|$ | $|AN|$ | $|AE|$ | Classes |
|---|---|---|---|---|
| AIDS | 2000 | 15.69 | 16.20 | 2 |
| PROTEINS | 1113 | 39.06 | 72.82 | 2 |
| NCI1 | 4110 | 29.87 | 32.30 | 2 |
| ENZYMES | 600 | 32.63 | 62.14 | 6 |
| DD | 1178 | 284.32 | 715.66 | 2 |

The datasets used in our experiments are all sourced from the publicly available `TUDataset` collection (Morris et al., 2020), available at https://chrsmrrs.github.io/datasets/. Table 1 shows the statistics of datasets. Here, $|G|$ denotes the number of graphs, $|AN|$ the average number of nodes, $|AE|$ the average number of edges, and *Classes* the number of target classes for each dataset.

### 6.1 ROBUSTNESS OF FSF

To assess FSF robustness, we perturb graphs by randomly adding or removing edges and measure changes via the bottleneck distance between the original and perturbed persistence diagrams. Experiments consider $H_1$ and $H_2$ on PROTEINS, NCI, and AIDS, with perturbations of edge removal (R) or addition (A) at ratios 0.05 and 0.1 of total edges.

The results (see Appendix A.5) demonstrate that the FSF is highly stable under both edge removal and addition, reflecting its robustness to structural noise. This robustness arises because the filtration is constructed from frequent subgraph patterns mined from the dataset, which capture stable and recurring topological structures.

## 6.2 DISCRIMINABILITY POWER OF FPH FEATURES

**FPH-ML performance.** We evaluate the performance of FPH-ML to assess the discriminability power of FPH features. A Support vector classifier (SVC) with a linear or Radial basis function (RBF) kernel is used. For each dataset, we use 10-fold evaluation and report the average accuracy and corresponding standard deviation. We compare FPH against two baselines: (i) degree-based PH (DPH) and (ii) the Weisfeiler–Lehman (WL) kernel. Additionally, comparing with random-label experiments (FPH-RL and DPH-RL) further confirms that the performance is not due to chance or label distributions.

Table 2: Classification accuracy (%) on benchmark datasets using SVC. A **bold** value indicates the best performance for each dataset.

| Methods | Datasets | | | | |
|---|---|---|---|---|---|
| | PROTEINS | AIDS | NCI1 | DD | ENZYMES |
| FPH | **74.31 ± 4.78** | **99.65 ± 0.59** | 70.00 ± 1.90 | **76.65 ± 3.21** | 36.67 ± 5.37 |
| DPH | 69.90 ± 4.89 | 97.50 ± 1.20 | 61.71 ± 1.59 | 74.53 ± 2.84 | 20.83 ± 5.39 |
| WL | 72.04 ± 3.36 | 98.30 ± 0.75 | **74.52 ± 2.85** | 76.12 ± 2.63 | **51.50 ± 6.34** |
| FPH-RL | 50.03 ± 4.54 | 51.06 ± 5.82 | 46.01 ± 3.72 | 43.07 ± 3.93 | 17.74 ± 3.72 |
| DPH-RL | 48.05 ± 3.62 | 52.01 ± 4.76 | 48.08 ± 5.83 | 47.02 ± 3.66 | 16.03 ± 4.04 |

Table 2 presents the results. FPH demonstrates superior performance compared to DPH across all datasets, achieving the best accuracy in PROTEINS, AIDS, and DD. For instance, on the PROTEINS dataset, FPH reaches 74.31%, compared to 69.90% with DPH. WL is competitive and achieves the highest performance on NCI1 (74.52%) and ENZYMES (51.50%), surpassing FPH. The random-label baselines yield drastic performance drops for both FPH and DPH, with accuracies around 16% for 6-class classification while 50% for binary classification, which is close to random guessing. These results show that the discriminative power of FPH-ML is not due to chance or label distribution but the inherent strength of FPH features in graph classification tasks.

**Contribution of different homology dimensions.** To further assess the power of FPH and the contribution of different homology dimensions, we set $k = 5$, which yields $H_3$ features that can potentially be killed. We evaluate six ablated variants of FPH-ML: (i) combining $H_0$–$H_2$ ($H_{0-2}$), (ii) combining $H_0$–$H_3$ ($H_{0-3}$), and (iii) using each of $H_0$, $H_1$, $H_2$, or $H_3$ individually.

Table 3: Classification accuracy (%) on benchmark datasets using SVC. Gray cells and **bold** values denote the best performance of combined FPH and individual dimensions, respectively.

| FPH | Datasets | | | | |
|---|---|---|---|---|---|
| | PROTEINS | AIDS | NCI1 | DD | ENZYMES |
| $H_{0-2}$ | 74.31 ± 4.78 | 99.65 ± 0.59 | 70.00 ± 1.90 | 76.65 ± 3.21 | 36.67 ± 5.37 |
| $H_{0-3}$ | 73.85 ± 3.56 | 99.60 ± 0.62 | 69.83 ± 2.17 | 76.23 ± 2.05 | 39.50 ± 5.37 |
| $H_0$ | 73.05 ± 4.36 | **99.60 ± 0.70** | **66.69 ± 1.37** | 75.04 ± 2.83 | 27.83 ± 4.60 |
| $H_1$ | **73.23 ± 3.57** | 90.10 ± 1.83 | 65.84 ± 1.92 | **76.23 ± 2.17** | 27.33 ± 6.37 |
| $H_2$ | 66.13 ± 4.79 | 80.00 ± 0.00 | 60.29 ± 1.81 | 74.79 ± 2.61 | 29.93 ± 5.49 |
| $H_3$ | 64.96 ± 4.63 | 80.25 ± 1.33 | 55.65 ± 1.71 | 74.02 ± 3.35 | **32.83 ± 3.80** |

Results in Table 3 show that $H_{0-2}$ slightly outperforms $H_{0-3}$ on most datasets except ENZYMES, and reveal that different datasets rely on different topological characteristics. For instance, on AIDS and NCI1, $H_0$ dominates, suggesting the importance of connected components, while on PROTEINS and DD, $H_1$ achieves the best accuracy, highlighting the role of cycle-related features. In contrast, ENZYMES benefits most from $H_3$, corresponding to tetrahedral structures.

**Discussion.** Since ENZYMES benefits most from higher-order homology, we conducted an additional experiment with $H_4$ and $H_{0-4}$, achieving 29.33% and **40.83%**, respectively. This reveals

that ENZYMES is closely related to higher-order topological structures. However, increasing $k$ will raise the computational cost of both FSM and PH calculation, and $H_{0-2}$ is generally sufficient for most datasets. Therefore, we adopt $H_{0-2}$ as the final FPH configuration. These results provide a clearer understanding of the role of topological features in graph classification, enhancing the interpretability.

### 6.3 FPH-GNN PERFORMANCE

In this experiment, we evaluate the performance of the proposed FPH-GNN on five widely used benchmark datasets. The baselines include classical GNN models: GCN, GIN, GraphSAGE, GAT, a pooling-based method Top-$k$Pool, and a persistent homology-based GNN model RePHINE (Immonen et al., 2023). The FPH features are integrated as a plug-in to the GNN models to enhance topology awareness. We perform 10-fold cross-validation, and Table 4 shows the mean classification accuracy with standard deviation.

Table 4: Classification accuracy (%) on benchmark datasets using GNN-based models. A **bold** value indicates the best performance for each dataset. Gray background indicates better performance between the model and its corresponding baseline.

| Methods | Datasets | | | | |
|---|---|---|---|---|---|
| | PROTEINS | AIDS | NCI1 | DD | ENZYMES |
| FPH-GCN | $78.61 \pm 3.83$ | $\mathbf{99.65 \pm 0.50}$ | $78.66 \pm 2.36$ | $\mathbf{82.94 \pm 3.49}$ | $\mathbf{47.00 \pm 6.32}$ |
| GCN | $74.48 \pm 1.73$ | $99.25 \pm 0.63$ | $74.52 \pm 1.13$ | $75.66 \pm 2.36$ | $38.82 \pm 3.31$ |
| FPH-GIN | $\mathbf{78.80 \pm 3.89}$ | $99.65 \pm 0.63$ | $79.08 \pm 2.45$ | $81.92 \pm 2.51$ | $44.67 \pm 5.57$ |
| GIN | $76.16 \pm 2.76$ | $99.25 \pm 0.53$ | $75.52 \pm 2.23$ | $76.05 \pm 3.60$ | $42.15 \pm 3.63$ |
| GAT | $74.72 \pm 4.01$ | $99.00 \pm 0.75$ | $74.90 \pm 1.83$ | $77.30 \pm 3.68$ | $39.83 \pm 3.68$ |
| GraphSAGE | $74.01 \pm 4.27$ | $98.20 \pm 1.05$ | $74.73 \pm 1.63$ | $75.78 \pm 3.98$ | $37.93 \pm 3.78$ |
| Top-$k$Pool | $75.03 \pm 1.29$ | $97.25 \pm 0.45$ | $78.92 \pm 2.14$ | $76.95 \pm 2.09$ | $38.35 \pm 3.65$ |
| RePHINE | $72.32 \pm 1.89$ | – | $\mathbf{80.92 \pm 1.92}$ | – | – |

The experimental results demonstrate that FPH-GNN models consistently outperform their baseline methods across all benchmarks. For example, FPH-GIN achieves the best performance (78.80%) on PROTEINS, compared to 76.16% for baseline GIN. Furthermore, FPH-GCN attains the best classification accuracy on AIDS (99.65%), DD (82.94%), and ENZYMES (47.00%), surpassing all baselines. On the NCI1 dataset, however, RePHINE slightly outperforms our proposed method. FPH-GCN and FPH-GIN achieve 78.66% and 79.09%, respectively, while RePHINE attains the best performance with 80.92%. Furthermore, the ablation study is shown in Appendix A.6.

Overall, the experiment results reveal that incorporating FPH features provides complementary global topological information, which can enhance graph-level representation learning, leading to more discriminative performance compared to purely local message-passing approaches.

## 7 CONCLUSIONS

We propose a novel filtration, FSF, for computing persistent homology on graphs. It is the first filtration constructed from frequent subgraph patterns mined across the entire dataset, and we also provide the theoretical analysis of its properties. In particular, we show that persistence obtained from FSF reflects both the frequency of patterns and the topology of the graph. FSF enriches the set of PH filtrations and bridges FSM with TDA for graph classification. Beyond persistent homology itself, we propose two approaches, FPH-ML and FPH-GNN, to incorporate the proposed PH features into graph classification. Extensive experiments on multiple benchmark datasets show that our methods outperform kernel-based, GNN-based, and PH-based baselines in most cases, highlighting the benefit of injecting high-order topological features into graph classification models. The computational cost of FSM and PH is a limitation of our method, which would benefit from general improvements in these methods. As future work, we aim to develop more efficient algorithms for computing frequency-based PH, for example, through approximate FSM mining with guarantees of topological similarity to exact methods. We also plan to further study the stability of frequent subgraph-based filtrations.

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

# A    APPENDIX

## A.1    DEFINITIONS OF PERSISTENT HOMOLOGY

**Definition 5 ($k-$Chain Group)** *Given a simplicial complex $\mathcal{K}$, let $\mathcal{K}_k$ denote the set of all $k$-simplices in $\mathcal{K}$. The k-chain group is defined as:*

$$C_k(\mathcal{K}) = \left\{ \sum_i a_i \sigma_i \mid a_i \in \mathbb{Z}, \sigma_i \in \mathcal{K}_k \right\},$$

*which forms a free Abelian group generated by the $k$-dimensional simplices.*

**Definition 6 (Boundary Operator)** *The boundary operator $\partial_p : C_p \to C_{p-1}$ is defined as:*

$$\partial_k(\sigma) = \sum_{i=0}^{k} (-1)^i [v_0, \ldots, \hat{v_i}, \ldots, v_k],$$

*where $[v_0, \ldots, v_k]$ is a $k$-simplex, and $[v_0, \ldots, \hat{v_i}, \ldots, v_k]$ represents the $(k-1)$-simplex obtained by removing the $i$-th vertex. An important property of this operator is: $\partial_k \circ \partial_{k+1} = 0, \forall k$. That is, boundaries have no boundary, meaning that each $(k+1)$-dimensional boundary forms a $k$-dimensional closed chain.*

**Definition 7 (Chain Complex)** *The boundary operator connects chain groups $C_k$ into a chain complex, given by:*

$$\cdots \xrightarrow{\partial_{k+2}} C_{k+1} \xrightarrow{\partial_{k+1}} C_k \xrightarrow{\partial_k} C_{k-1} \xrightarrow{\partial_{k-1}} \cdots \xrightarrow{\partial_1} C_0 \xrightarrow{\partial_0} 0.$$

**Definition 8 ($k$-Boundary Group)** *Given a simplicial complex $\mathcal{K}$, the k-boundary group is defined as:*

$$B_k(\mathcal{K}) = \mathrm{Im}(\partial_{k+1}) = \{\partial_{k+1}(\sigma) \mid \sigma \in C_{k+1}\}.$$

**Definition 9 ($k$-Cycle Group)** *Given a simplicial complex $\mathcal{K}$, the $k$-cycle group is defined as:*

$$Z_k(\mathcal{K}) = \ker(\partial_k) = \{c \in C_k \mid \partial_k(c) = 0\}.$$

Since $B_k \subset Z_k$ and $Z_k \subset C_k$, this implies that $Z_k$ consists of $k$-cycles (i.e., $k$-closed chains), while $B_k$ consists of $k$-boundaries (i.e., $k$-closed chains that are also boundaries). The goal of homology is to distinguish nontrivial cycles from boundaries, leading to the definition of the homology group:

**Definition 10 (Homology Group)** *The $k$-th homology group of a simplicial complex is defined as:*

$$H_k(\mathcal{K}) = Z_k(\mathcal{K})/B_k(\mathcal{K}).$$

*Its rank is called the Betti number, denoted as $\beta_k$, which represents the number of $k$-dimensional nontrivial topological features:*

$$\beta_k = \dim H_k(\mathcal{K}).$$

**Definition 11 (Persistence Diagram)** *Given a filtered simplicial complex $\{K_t\}_{t \in \mathbb{R}}$, the $p$-dimensional persistent homology captures the birth and death times of $p$-dimensional topological features as the filtration parameter $t$ increases. The persistence diagram $\mathrm{PD}_p$ is a multiset of points in $\mathbb{R}^2$ of the form $(b, d)$, where $b$ and $d$ denote the birth time and death time, respectively, of a $p$-dimensional feature, with $b < d \leq \infty$. Features with infinite death time are often treated as points at infinity or capped at a fixed value for computational purposes.*

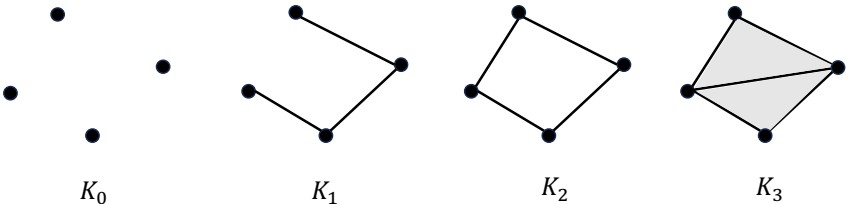

$K_0 \qquad K_1 \qquad K_2 \qquad K_3$

Figure 4: Example of filtration.

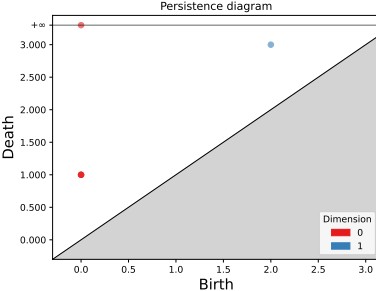

Figure 5: Example of persistence diagram.

For example, figure 4 illustrates an example of a filtration sequence, and figure 5 shows the persistent diagram. For $H_0$ (red points in the persistent diagram), there are four components at filtration 0, but three of them die at filtration 1. For $H_1$ (blue points in the persistent diagram), a cycle is born at filtration 2, and it dies at filtration 3 because two 2-simplices are added to kill it.

**Definition 12 (Bottleneck Distance)** *Let $D_1$ and $D_2$ be two persistence diagrams. The bottleneck distance between $D_1$ and $D_2$ is defined as*

$$d_B(D_1, D_2) = \inf_{\gamma} \sup_{x \in D_1} \|x - \gamma(x)\|_{\infty},$$

*where $\gamma$ ranges over all bijections between $D_1$ and $D_2$, and $\|\cdot\|_{\infty}$ denotes the $\ell_{\infty}$ norm. Intuitively, the bottleneck distance measures the largest shift needed to match the points of one diagram to the other.*

## A.2 PROOF OF PROPOSITIONS

**Proof of Proposition 1** For every frequent subgraph pattern $p$, if it is isomorphic with the subgraphs of $G$, it contributes vertex sets $T \subset I(V_p)$ of size at most $k$, which induces simplexes $\triangle(T)$ of dimension at most $k - 1$. Thus, for every simplex $\sigma \in \mathcal{K}_t$, the dimension $\dim(\sigma) \leq k - 1$. Let $C_p(\mathcal{K}_t)$ be the $p$-chain group. Then, we have $C_p(\mathcal{K}_t) = 0$ for all $p \geq k$. By the definition of $p$-th homology group $H_p(\mathcal{K}_t) = \ker(\partial_p)/\mathrm{im}(\partial_{p+1})$, where $\partial_p : C_p \to C_{p-1}$ is the boundary operator. Since $C_p = 0$ for all $p \geq k$, we have $H_p(\mathcal{K}_t) = 0$ for all $p \geq k$. Therefore, the maximum dimension for nontrivial persistent homology is $k - 1$.

Furthermore, for $p = k - 1$, the image of the boundary operator $\partial_k$ is 0 (as $C_k = 0$). Hence, $\mathrm{im}(\partial_k) = 0$. Then we have $H_{k-1} = \ker(\partial_{k-1})$. This implies that any $(k-1)$-hole, once formed, cannot be filled by higher-dimensional simplices. Therefore, such classes persist indefinitely in the filtration.

$\square$

**Example of non-vanishing**

As shown in Figure 6, we subdivide the edges $\{a, b\}, \{a, c\}, \{b, c\}$ by introducing the vertices $d, e, f$, respectively. If the pattern size is equal to 3, $\forall t \gg 1$, the complex $\mathcal{K}_t$ contains the following 2-simplices as well as their subsimplices: $\triangle abd, \triangle ace, \triangle ade, \triangle bcf, \triangle bdf,$ and $\triangle cef$.

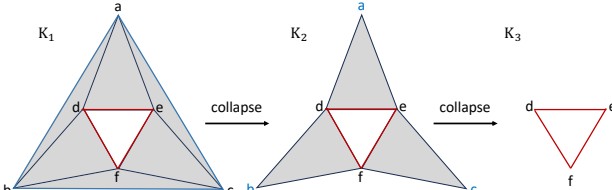

Figure 6: Example of non-vanishing.

Considering $H_1$, in the complex $\mathcal{K}_1$ the 1-simplex $\{a, b\}$ is only contained in the 2-simplex $\triangle abc$, hence it is a free face; the same holds for $\{a, c\}$ and $\{b, c\}$. By performing elementary collapses on these free faces, we obtain the reduced complex $\mathcal{K}_2$.

In $\mathcal{K}_2$, the vertices $a, b, c$ become free faces contained uniquely in the $\triangle ade, \triangle dbf,$ and $\triangle efc$, respectively. Collapsing them gives the final reduced complex $\mathcal{K}_3$.

Therefore, we conclude that

$$H_1(\mathcal{K}_1) \cong H_1(\mathcal{K}_3) \neq 0.$$

**Proof of Proposition 2** Let $G$ and $G'$ be two graphs such that $G \cong G'$. Let $\phi : V(G) \to V(G')$ be a vertex-label-preserving graph isomorphism. For any subgraph $S \subseteq G$, there exists an isomorphic subgraph $\phi(S) \subseteq G'$. This subgraph $\phi(S)$ induces the corresponding vertex set $V_{\phi(S)}$. Hence, under the same frequent subgraph threshold, the sets of frequent subgraph vertices coincide, i.e., $V_{S \subseteq G} = V_{\phi(S) \subseteq G'}$. Therefore, at each step of the filtration, an isomorphic simplicial complex is obtained. In particular, the filtration $\{\mathcal{K}'_t\}_{t \in \mathbb{R}_+}$ on $G'$ satisfies $\mathcal{K}_t \cong \mathcal{K}'_t$ for all $t$. Thus, the persistence diagrams are identical: $D_G = D_{G'}$. $\square$

**Proof of Proposition 3** As $t$ increases, the frequency threshold $\theta = \frac{1}{t}$ decreases, allowing more subgraph patterns to be considered frequent:

$$t_1 < t_2 \quad \Rightarrow \quad \frac{1}{t_1} > \frac{1}{t_2} \quad \Rightarrow \quad P_{t_1} \subseteq P_{t_2},$$

where $P_t$ denotes the set of subgraph patterns frequent at threshold $1/t$.

For each frequent subgraph pattern $p \in P_t$, let $\mathrm{Emb}(p, G)$ denote the set of all isomorphic embeddings of $p$ into $G$. Each such embedding induces a vertex set $T \subseteq V(G)$, which defines a simplex $\triangle(T)$ in the simplicial complex. Thus, as $t$ increases, more subgraph patterns and their embeddings are included, contributing additional simplices to the complex:

$$\mathcal{K}_{t_1} \subseteq \mathcal{K}_{t_2}, \quad \text{for all } t_1 < t_2.$$

Hence, the filtration is monotonic in $t$. $\qquad\square$

## A.3 FREQUENT SUBGRAPH-BASED FILTRATION

Figure 7 illustrates the overall process of the filtration construction.

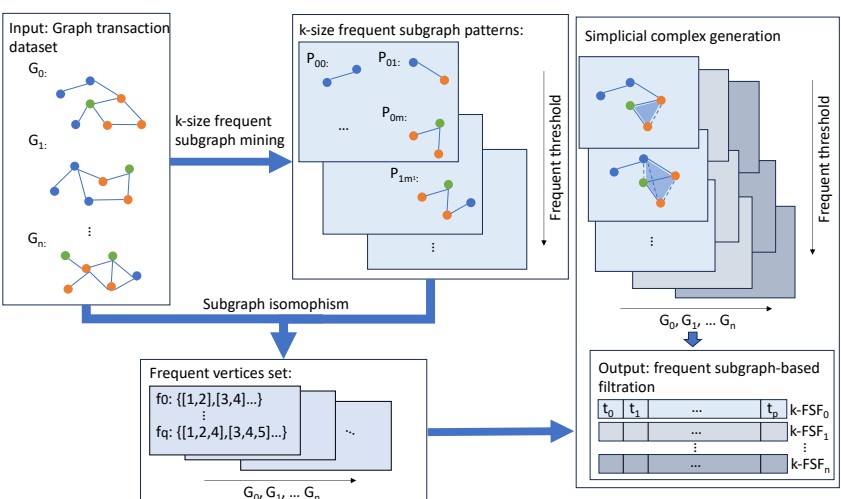

Figure 7: k-FSF construction.

FPH features are computed from the following statistical descriptors based on the lifetimes:

i) Mean lifetime $\mu_i^k = \frac{1}{n_k} \sum_{j=1}^{n_k} \ell_j^k$;

ii) Maximum lifetime $L_{\max,i}^k = \max_j \ell_j^k$;

iii) Minimum lifetime $L_{\min,i}^k = \max_j \ell_j^k$;

iv) Median lifetime $\tilde{L}_i^k = \mathrm{median}\{\ell_j^k\}_{j=1}^{n_k}$;

v) Standard deviation $\sigma_i^k = \sqrt{\frac{1}{n_k} \sum_{j=1}^{n_k} (\ell_j^k - \mu_i^k)^2}$;

vi) Betti number $\beta_i^k = n_k$;

vii) Entropy $E_i^k = \begin{cases} -\sum_{j=1}^{n_k} p_j^k \log_2 p_j^k, & \text{if } \sum_j \ell_j^k > 0, \\ 0, & \text{otherwise.} \end{cases}$

## A.4 DETAILS OF GNN LAYERS

**GNN layers:** Then, we use $L$ layers of GNN to propagate both structural and topological information:

$$H_i^{(l)} = \mathrm{GNN}^{(l)}(A_i', H_i^{(l-1)}) \quad \text{for } l = 1, \dots, L,$$

where $A_i'$ is the new adjacency including the FPH token, and $H_i^{(l)} \in \mathbb{R}^{n_i \times d}$ denotes the hidden node representations.

Finally, to guarantee that the encoded FPH features are not overshadowed or diminished after multiple layers of message passing, we use a residual skip connection mechanism from the initial embedding $H_i^{(0)}$ to the final representation $H_i^{(L)}$:

$$H_i^* = \mathrm{ReLU}\big(H_i^{(0)} + H_i^{(L)}\big).$$

## A.5 ROBUSTNESS EXPERIMENT

Figure 8 shows the persistent diagram of the original graph and the perturbed graphs across these three datasets, and Table 5 shows the bottleneck distance results. As shown in table 5, the bottleneck distances for all datasets and perturbation settings remain extremely small, typically on the order of $10^{-4}$ to $10^{-2}$. For example, for PROTEINS, the distances for $H_1$ remain below $1.1 \times 10^{-3}$ even with a 0.1 perturbation ratio, and for $H_2$ they remain under $2 \times 10^{-3}$. Across all three datasets, increasing the perturbation ratio from 0.05 to 0.1 leads to only minor increases in bottleneck distance. For instance, in NCI1, $H_1$ exhibits an increase of only about $2 \times 10^{-3}$ for edge removal, while the increase for $H_2$ is about $5 \times 10^{-3}$. The performance shows a similar trend for edge addition.

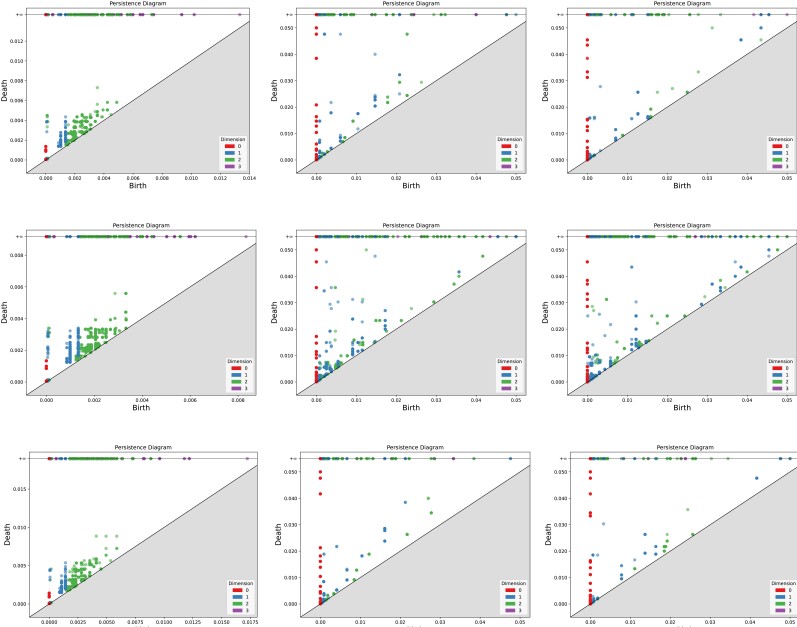

Figure 8: Persistence diagrams for three datasets under different perturbations. Row 1: Original graphs. Row 2: Graphs after adding 10% of edges. Row 3: Graphs after removing 10% of edges. Column 1: PROTEINS. Column 2: AIDS. Column 3: NCI1.

Table 5: Bottleneck distances for $H_1$ and $H_2$.

| Dataset | H-dim | R_0.05 | R_0.1 | A_0.5 | A_0.1 |
|---------|-------|--------|-------|-------|-------|
| PROTEINS | $H_1$ | $9.797 \times 10^{-4}$ | $1.029 \times 10^{-3}$ | $9.629 \times 10^{-4}$ | $9.629 \times 10^{-4}$ |
|          | $H_2$ | $1.604 \times 10^{-3}$ | $1.953 \times 10^{-3}$ | $9.573 \times 10^{-4}$ | $1.719 \times 10^{-3}$ |
| NCI1 | $H_1$ | $5.562 \times 10^{-3}$ | $7.221 \times 10^{-3}$ | $1.834 \times 10^{-2}$ | $1.936 \times 10^{-2}$ |
|      | $H_2$ | $4.819 \times 10^{-3}$ | $9.402 \times 10^{-3}$ | $2.230 \times 10^{-2}$ | $2.276 \times 10^{-2}$ |
| AIDS | $H_1$ | $5.231 \times 10^{-3}$ | $1.907 \times 10^{-2}$ | $1.435 \times 10^{-2}$ | $1.436 \times 10^{-2}$ |
|      | $H_2$ | $1.245 \times 10^{-2}$ | $1.246 \times 10^{-2}$ | $1.351 \times 10^{-2}$ | $1.355 \times 10^{-2}$ |

## A.6 ABLATION STUDY OF FPH-GNN

To assess the contribution of different components in our proposed FPH-GNN models, we conduct comprehensive ablation studies on PROTEINS, AIDS, NCI1, DD, and ENZYMES, considering both FPH-GCN and FPH-GIN models. We evaluate three ablated versions of our model: (i) No Gate (NG): removing the fusion gate $\alpha$ and directly summing node and token representations; (ii) Full Connection (FC): replacing the top-$k$ token injection strategy with full connections from the global

topology token to all nodes; (iii) No Residual skip connection (NRsc): eliminating the residual skip connection in the last layer.

Table 6: Ablation study for FPH-GNN.

| Methods | Datasets | | | | |
|---|---|---|---|---|---|
| | PROTEINS | AIDS | NCI1 | DD | ENZYMES |
| FPH-GCN | **78.61 ± 3.83** | **99.65 ± 0.50** | **78.66 ± 2.36** | **82.94 ± 3.49** | **47.00 ± 6.32** |
| FPH-GCN-NG | 78.08 ± 4.65 | 99.60 ± 0.62 | 77.92 ± 4.62 | 81.36 ± 2.42 | 45.67 ± 5.88 |
| FPH-GCN-FC | 78.52 ± 3.95 | 99.60 ± 0.62 | 77.79 ± 2.55 | 81.49 ± 2.99 | 42.33 ± 5.73 |
| FPH-GCN-NRsc | 78.35 ± 3.31 | 99.60 ± 0.62 | 78.64 ±1.70 | 82.30 ± 3.04 | 45.17 ± 4.91 |
| GCN | 74.48 ± 1.73 | 99.25 ± 0.63 | 74.52 ± 1.13 | 75.66 ± 2.36 | 38.82 ± 3.31 |
| FPH-GIN | **78.80 ± 3.89** | **99.65 ± 0.63** | **79.08 ± 2.45** | **81.92 ± 2.51** | **44.67 ± 5.57** |
| FPH-GIN-NG | 76.98 ± 4.58 | 99.55 ± 0.60 | 78.25 ± 2.80 | 80.90 ± 2.98 | 43.50 ± 7.32 |
| FPH-GIN-FC | 78.38 ± 4.78 | 99.55 ± 0.61 | 78.22 ± 1.16 | 77.93 ± 1.96 | 40.67 ± 7.23 |
| FPH-GIN-NRsc | 77.98 ± 5.35 | 99.55 ± 0.65 | 78.35 ± 2.25 | 79.23 ± 2.98 | 43.25 ± 5.58 |
| GIN | 76.16 ± 2.76 | 99.25 ± 0.53 | 75.52 ± 2.23 | 76.05 ± 3.60 | 42.15 ± 3.63 |

As shown in Table 6, removing any of these modules consistently reduces performance compared to the full FPH-GNN. These results highlight the importance of each design choice. For instance, directly summing graph and token embeddings (NG) leads to performance drops, since it removes the adaptive fusion of global and local features. Similarly, fully connecting the global topology token (FC) may introduce noise, thereby reducing the informative role of FPH features. The residual connection (NRsc) also proves essential, as it enhances the influence of FPH features after several GNN layers.

A noteworthy observation is that ablating these modules degrades performance less severely than removing the FPH features themselves, demonstrating that FPH features are the key factor driving the improvement of FPH-GNN over plain GNN baselines.

