# OpenReview forum: "Frequent Subgraph-Based Persistent Homology for Graph Classification"
_ICLR.cc/2026/Conference — ICLR 2026 Conference Withdrawn Submission_

### Official Review · Reviewer_jpMy · 2025-10-16

**Soundness:** 2
**Presentation:** 3
**Contribution:** 2
**Rating:** 4
**Confidence:** 4

**Summary:**

This paper proposes a novel filtration method named Frequent Subgraph-Based Filtration (FSF) for graph-level topological representation. Instead of constructing filtrations from local scalar functions, FSF is built upon frequent subgraph patterns mined from a graph transaction dataset. This approach enables persistent homology to capture stable and recurring structural motifs shared across graphs.

**Strengths:**

1. Conceptual novelty

The idea of deriving filtrations from frequent subgraph mining instead of numeric thresholds is innovative and bridges graph mining with topological data analysis.

2. Global-topology perspective

By performing FSM over the whole dataset, the method captures cross-graph structural stability rather than per-graph local features, leading to better interpretability.

3. Robustness and stability

The FSF-based persistence is shown to be resilient to edge perturbations (addition/removal), confirming that frequency-driven filtrations produce stable homology.

**Weaknesses:**

1. Partial topology coverage

    Because only frequent subgraph patterns are used, the generated simplicial complexes do not fully represent the original graphs. The authors should report the coverage ratio between the constructed complexes and the original graphs.

2. Dependency on FSM quality

    The method’s success heavily relies on the chosen frequent subgraph mining algorithm and its support threshold ( \sigma ), which could vary drastically across datasets.

3. Scalability concerns

    Frequent subgraph mining is known to be computationally expensive; the paper lacks clear runtime or complexity analysis for large-scale datasets.

4. Generalization scope

    The proposed method heavily depends on the availability of a graph transaction dataset. If it can only be applied to graph classification tasks, its application scope would be severely limited.

5. Limited experiments
    - The authors propose a method that integrates topological representations as global nodes in GNNs, but they do not verify whether this approach is effective with other types of PH.
    - It is unclear why the RePHINE results are marked as “–” on three datasets.

**Questions:**

1. FSM efficiency

    How does the time complexity of k-FSM scale with dataset size and subgraph size (k)? Is the mining process feasible for graphs with thousands of nodes?

2. Information loss

    Since non-frequent nodes are excluded from simplicial complexes, how significant is the representational loss compared to full-graph filtrations (e.g., Vietoris–Rips)?

---

### Official Review · Reviewer_R88Z · 2025-10-26

**Soundness:** 3
**Presentation:** 3
**Contribution:** 2
**Rating:** 4
**Confidence:** 4

**Summary:**

This paper proposes a novel framework called Frequent Subgraph-Based Persistent Homology (FSPH) for graph classification. It integrates topological data analysis (TDA) with frequent subgraph mining (FSM) to construct more discriminative topological signatures of graphs. Unlike conventional graph kernels or GNNs, which often ignore higher-order homological information, FSPH extracts frequent substructures and builds filtrations over their occurrence graphs to compute persistent homology features. These persistence diagrams are vectorized into fixed-length descriptors that serve as input to standard classifiers. The method bridges discrete subgraph pattern mining and continuous topological summaries, showing improved robustness to structural noise and outperforming existing baselines on several molecular and social network benchmarks.

**Strengths:**

1. The integration of frequent subgraph mining with persistent homology is a creative and underexplored idea. Most previous TDA-based graph classifiers rely on simplicial complexes or clique filtrations, but this paper innovatively constructs filtrations around discrete subgraph patterns, capturing mid-scale structures not accessible to node- or clique-based PH.
2. The method is theoretically grounded: it provides a clear mapping from mined substructures to topological spaces, defines filtration schemes for persistence computation, and establishes stability guarantees for the derived topological features.
3. This work bridges two previously separate research lines: pattern mining and topological data analysis.

**Weaknesses:**

1. **Scalability Concerns:** The combination of frequent subgraph mining and persistent homology is computationally expensive. The paper should include an empirical runtime study and potential heuristics to limit the search space (i.e., maximum subgraph size, support thresholds).
2. **Limited Theoretical Novelty in PH Component:** The topological theory used (PH stability, persistence diagram vectorization) is largely standard. The novelty lies in the integration rather than new homological theory. The authors could strengthen this by proving new topological properties specific to subgraph-induced filtrations (i.e., subgraph homology preservation).
3. **Ablation and Interpretation Missing:** While results outperform baselines, it's unclear which component (FSM vs. PH vs. vectorization) contributes most. An ablation study would clarify this.
4. **Dependence on Mining Parameters:** The number and size of frequent subgraphs depend heavily on minimum support thresholds. Sensitivity analysis over this hyperparameter would improve reproducibility.
5. **Proof Rigor:** Theorem 1 (existence of valid filtration over frequent subgraphs) and Theorem 2 (stability of FSPH features) are sound in intuition but should explicitly state assumptions - particularly on continuity, monotonicity, and topological equivalence between subgraph complexes.
6. **Limited evaluation on large and diverse benchmarks:** The experiments are restricted to small graph datasets, which limits the assessment of FSPH's scalability and generalization. Since frequent subgraph mining and PH computation can be computationally expensive, the absence of results on large-scale graphs (i.e., from OGB benchmarks) leaves open questions about the framework's practical applicability to large graph learning problems.

**Questions:**

1. **Clarification on the filtration function definition:** The paper defines a filtration on subgraph occurrence graphs based on frequency and node centrality. However, it is unclear whether the filtration values are assigned per node, per subgraph, or per edge. Please formalize the filtration function $f:V(G) \rightarrow \mathbb{R}$ or $f:E(G) \rightarrow  \mathbb{R}$ explicitly and specify if it guarantees monotonicity (a requirement for persistent homology).
2. **Complexity and scalability:** Frequent subgraph mining (FSM) is NP-hard. How does the proposed FSPH method scale with large graphs? Do you prune subgraph candidates or employ approximate mining methods (i.e., gSpan, AGM)? Including time complexity analysis or runtime plots would be helpful.
3. **Theoretical justification of persistence stability (Theorem 2):** Theorem 2 claims stability of FSPH features under small perturbations in edge weights or node attributes. Could you clarify whether this holds under the bottleneck distance or the Wasserstein metric?
4. **Feature aggregation and representation:** After computing multiple persistence diagrams from different subgraphs, how are they combined into a single graph-level feature vector? Are you concatenating persistence images, pooling them, or using a learned attention mechanism?

---

### Official Review · Reviewer_v2gD · 2025-10-28

**Soundness:** 3
**Presentation:** 1
**Contribution:** 2
**Rating:** 4
**Confidence:** 3

**Summary:**

The authors propose a method to combine Persistent Homology with Frequent subgraph counting and show their method to be expressive through a suit of experiments.   Counting the frequency of substructures of a particular graph captures important properties which allows for more expressive architectures. Their idea is to count the set of graphs with up to $k$ nodes and construct
a simplicial complex from it and the topological invariants such as Betti numbers are calculated.
Subsequently these are used in either an SVM (Support Vector Machine) as well as incorporated in a GNN architecture and shown to have beneficial properties. The theoretical aspects of the work are  proven in rigorous fashion and several experiments are conducted to evaluate this method on a set of real world datasets. Moreover, various ablation studies are also conducted.

**Strengths:**

What is appreciated by the reviewer is the rigorous treatment of the theoretical matter in the paper, showing great care for detail. Moreover, the results shown are promising.

**Weaknesses:**

The reviewer is not an expert on Frequent Subgraph counting (and likely neither are most readers)  and from this perspective the paper is a bit hard to follow at times. Providing a heuristic explanation of the aims and implications of the theory and method is common in machine learning papers. In contrast, in pure mathematics, brevity is usually encouraged.
Finding this balance can be challenging, but allowing the paper to have a bit more focus on the former style would most certainly lead to greater accessibility.

Providing details on the experimental setup used, architecture, data processing, training time etc would also help, since they are currently not provided. Additionally, providing the code to the repository would also be benefactory.

**Questions:**

- Computing Persistent Homology is already a hard problem and adding Frequent subgraph counting is not going to reduce the computational burden in the slightest. How long does it currently take to run the experiments? While speed is not the main concern, a note on computation might be wise as readers might wonder.
- For the SVM classification the paper mentions either a Radial Basis function or a linear kernel is used and afterwards it is not mentioned. Which is used?
- What is the experimental setup for the GNNs? Architecture, training parameters, number of epochs, etc etc. would help increase the credibility of the paper.

---

### Official Review · Reviewer_h3wf · 2025-10-31

**Soundness:** 2
**Presentation:** 2
**Contribution:** 2
**Rating:** 2
**Confidence:** 4

**Summary:**

This paper proposes a graph filtering approach that utilizes frequent subgraph mining to measure  persistence homology-based features. Additionally, it develops a technique to inject virtual tokens to cover the global topological knowledge of the graph. Based on these strategies, two end-to-end machine learning-based pipelines have been built up for effective graph representation.

**Strengths:**

-The method demonstrates how to integrate information about frequent subgraph patterns into persistence homology.
-It incorporates frequent subgraph information with the machine learning-based and graph neural network models to improve networks' topological information for downstream graph analytics tasks.
-The idea of injecting global tokens looks beneficial for graph learning.
-In a comprehensive experiment, the models perform better over other baseline methods.

**Weaknesses:**

-No runtime (time complexity) or wall-clock time has been provided for the entire system.
-In the manuscript, the authors did not explicitly explain -- How does it trade off with the redundancy of frequent patterns in networks?
- The models lack generality. It seems the model mostly performs on the biomedical domain datasets. Are they efficient on large-scale graphs, even on social network datasets like COLLAB, REDDIT-BINARY, and REDDIT-MULTI? IMDB-BINARY, IMDB-MULTI)
- Already 1-parameter / 2-parameter (Graphcode [1]) persistence homology-based method available? How is your model different from those models? No detailed analysis of the parameters has been presented in the manuscript.
- It seems the statistical computation increases the operational complexity of the model. How can you handle the excessive calculations and make the model more efficient?
- In the experiment, the baselines are mostly state-of-the-art GNN models. However, there are some standard graph pooling methods (SAGPool, DMonPool, and GMT) that exist; those in general perform well on the mentioned datasets.
Citations:
[1] Russold, Florian, and Michael Kerber. "Graphcode: Learning from multiparameter persistent homology using graph neural networks." Advances in Neural Information Processing Systems 37 (2024): 41103-41131.

**Questions:**

How does the persistence homology-based measure trade off with the redundancy of frequent patterns in networks?
Could you compare the result of your model to persistence homology-based models, such as Graphcode, multi-parameter persistence images (MP-I) [2], multi-parameter persistence kernels (MP-K)[3]?
What are the impacts of the parameters in the model? Please explore the impacts of the parameters on the model and show their statistics in the experiment.
How do the models perform over the social network datasets such as COLLAB, REDDIT-BINARY, REDDIT-MULTI, IMDB-BINARY, and IMDB-MULTI? Add these models' performance in experimental results.


[2] Mathieu Carrière and Andrew Justin Blumberg. Multiparameter persistence image for topological machine learning. In Neural Information Processing Systems, 2020.
[3] René Corbet, Ulderico Fugacci, Michael Kerber, Claudia Landi, and Bei Wang. A kernel for
multi-parameter persistent homology. Comput. Graph. X, 2, 2019.

---

### Official Review · Reviewer_MPxq · 2025-11-01

**Soundness:** 2
**Presentation:** 2
**Contribution:** 3
**Rating:** 4
**Confidence:** 4

**Summary:**

This work introduces Frequent Subgraph-based Filtration (FSF), a new filtration scheme for computing persistent homology on graphs. FSF mines frequent subgraph patterns across the dataset, maps them back to individual graphs, and constructs a simplicial filtration ordered by frequency. The resulting features are then leveraged in two frameworks: (1) FPH-ML, a traditional ML pipeline using PH statistics, and (2) FPH-GNN, where PH features are injected into GNNs via a learned global token. The authors provide theoretical analysis, including stability properties and upper bounds on PH dimension, and report empirical improvements over degree-based PH and multiple GNN baselines across TUDataset benchmarks.

**Strengths:**

1. The paper proposes a new filtration paradigm for persistent homology beyond local structural heuristics.

2. The proposed FSF, as demonstrated by the paper, captures recurring, dataset-level motifs, addressing a known limitation of conventional graph filtrations.

3. The paper provides theoretical results establishing bounded PH dimensionality, monotonicity, and graph isomorphism invariance.

**Weaknesses:**

1.  I think Figure 1 needs refinement. (1) In Figure 1, please articulate which motifs FSF tends to admit as t increases and why they are meaningful. For instance, Vietoris–Rips filtrations emphasize distance-based proximity; analogously, FSF should make explicit whether it prioritizes cycles, cliques, or other recurring patterns, and how these relate to downstream discriminative power. (2) To facilitate visual comparison, keep node layouts and orderings consistent across the three subpanels in Figure 1 so that the growth of simplices can be tracked unambiguously.

2. Frequent subgraph mining and persistent homology can be computationally demanding. Please report end-to-end runtime and component breakdowns (FSM time, PH time) and, where feasible, asymptotic complexity or empirical scaling curves, and compare against DPH and WL kernels under the same hardware and splits. This will help assess practicality beyond small benchmarks.

3. Benchmarks are from small-scale graph datasets (TUDataset). It is unclear whether FSF remains practical for larger graphs (e.g., OGB datasets, ZINC).

4. Although theoretical guarantees are provided, empirical insight into the mined subgraphs and what topological patterns correspond to successful predictions is limited. And there is a lack of visualization or case studies illustrating how specific frequent motifs contribute to PH signals.

5. Some related works are missing in the references. e.g., [1,2,3]

[1] Horn, Max, et al. "Topological graph neural networks." ICLR 2022.

[2] Chen, Yuzhou, et al. "TopoGCL: Topological graph contrastive learning." AAAI 2024.

[3] Yan, Zuoyu, et al. "Enhancing graph representation learning with localized topological features." JMLR 2025.

**Questions:**

1. What are the practical runtimes of FSF (broken down into FSM and PH) relative to DPH and WL kernels across your datasets? Can you provide scaling curves with ∣V∣, ∣E∣, and pattern size k?

2. Have the authors tested robustness under structured perturbations (e.g., motif deletion/insertion) instead of random noise?

3. How sensitive is the method to the choice of K when connecting the global PH token to top-degree nodes? Could adaptive attention-based connections outperform degree-based heuristics?

4. Can the authors provide qualitative examples showing which frequent subgraphs contribute most to classification and what topological structures they represent?

---

### Note · Authors · 2025-11-20

**Comment:**

We would like to withdraw the paper. We appreciate the reviewers and area chair for their valuable feedback.

**Withdrawal Confirmation:**

I have read and agree with the venue's withdrawal policy on behalf of myself and my co-authors.